# Adverse Events and Associated Economic Burden of COVID-19 Vaccination in Queensland, Australia: Findings from the Cross-Sectional QoVAX-Statewide Study

**DOI:** 10.3390/vaccines13070712

**Published:** 2025-06-30

**Authors:** Qing Xia, Kerry-Ann F. O’Grady, Peter Vardon, Selina Ward, Rebecca Gregory, Janet Davies, Hannah E. Carter

**Affiliations:** 1Australian Centre for Health Services Innovation and Centre for Healthcare Transformation, School of Public Health & Social Work, Faculty of Health, Queensland University of Technology, Brisbane, QLD 4059, Australia; kerryann.ogrady@qut.edu.au (K.-A.F.O.); hannah.carter@qut.edu.au (H.E.C.); 2Metro North Health, Brisbane, QLD 4006, Australia; peter.vardon@health.qld.gov.au (P.V.); selina.ward@student.uq.edu.au (S.W.); rebecca.gregory@health.qld.gov.au (R.G.); j36.davies@qut.edu.au (J.D.); 3Centre for Immunology and Infection Control, School of Biomedical Sciences, Faculty of Health, Queensland University of Technology, Brisbane, QLD 4059, Australia

**Keywords:** COVID-19, vaccination, adverse events, economic analysis, productivity loss

## Abstract

**Background/Objectives**: The economic impact of adverse events following COVID-19 immunisation (AEFIs) in Australia is underexplored. This study aimed to assess the economic burden of AEFIs on both healthcare systems and societal productivity. **Methods**: A cross-sectional survey was conducted in Queensland residents aged ≥18 years who had received at least one dose of a COVID-19 vaccine in the preceding 12 months. Overall, 6964 participants were recruited from July to September 2022 via email and broad social media campaigns. The survey collected data on the incidence, type and duration of AEFIs; healthcare utilisation; and work-related absenteeism. Healthcare costs were estimated using national healthcare reimbursement data, and productivity costs were estimated using Australian Bureau of Statistics Average Weekly Earnings. **Results**: Of the 6797 eligible respondents (predominantly female [62%]; median age: 52 years), AEFIs were reported by 53.4%, 44.1%, 40.7%, and 40.9% following doses 1 to 4, respectively. Pain and tenderness were predominant local AEFIs, while tiredness and headaches were the most frequent systemic AEFIs, generally resolving within three days. Relatively few participants reporting AEFIs consulted medical professionals: 7.0%, 7.3%, 5.1%, and 1.9% following each dose, respectively. The mean healthcare cost per person reporting AEFIs was AUD 24, AUD 88, AUD 22, and AUD 4 following each respective dose. Work absenteeism was recorded in 16.5%, 18.2%, 15.2%, and 11.2% following each dose with mean absenteeism days per person of 4.7, 7.4, 3.6 and 2.1, respectively, and mean productivity costs per person reporting AEFIs amounting to AUD 1494, AUD 2388, AUD 1136, and AUD 690, respectively. **Conclusions**: Participants reported mostly mild AEFIs with only a small proportion of individuals seeking medical services. Productivity costs attributable to these AEFIs exceeded direct healthcare expenses incurred.

## 1. Introduction

The COVID-19 pandemic had far-reaching health and economic impacts globally, prompting the implementation of comprehensive vaccination programs as a critical measure to mitigate the spread of the virus [1]. Australia launched its national COVID-19 immunisation campaign on 22 February 2021 with the ambitious goal of vaccinating the entire population by the end of 2021 [2]. Approximately 92% of people in the Australian state of Queensland had received the primary course of COVID-19 vaccination by December 2021.

COVID-19 vaccines have demonstrated safety and efficacy with rare major case fatalities [3]; however, some individuals may experience adverse events following immunisation (AEFIs) [4]. According to the Australian Centre for Disease Control [5], the annual AEFI reporting rate was 271.4 per 100,000 COVID-19 vaccine doses administered to people aged ≥ 12 years, and the most frequently reported symptoms include fatigue, headache, myalgia, and pyrexia. These events are typically transient and self-limiting, and they are considered a normal immune response to vaccination [4].

While extensive research has focused on the safety and efficacy of COVID-19 vaccines [6,7], there is a notable lack of research examining the potential economic consequences of AEFIs on health systems and society. International research has mainly focused on hospitalisations or work absences of frontline healthcare providers [8,9,10,11]. These studies indicated that there is a risk of additional staff shortages due to post-vaccination inability to work with estimates varying across publications 1.6–13.4% after the first dose and 6.1–28.4% after the second dose. To date, two Australian studies found that 0.9–1.4% of vaccinated individuals required medical attention for AEFIs, and 4.2–20.3% reported missed work, study or routine activities (Appendix A) [12,13]. However, the cumulative impact of these AEFIs on both individuals and society, particularly across multiple doses administered over a relatively short period, warrants further investigation. In particular, the economic costs to society and the health system associated with these adverse events have not been quantified.

This study aims to address these research gaps by investigating the AEFI profiles (type, frequency and duration) and associated economic burden across four vaccine doses, using self-report questionnaires as part of the Queensland COVID-19 Vaccination Program Statewide Study (QoVAX Statewide Study) [14,15]. By quantifying economic impacts, this study will contribute to evidence-based decision making and resource allocation within public health systems, facilitating the effective implementation and optimisation of COVID-19 vaccination programs. Understanding the economic implications of AEFIs may also aid in developing strategies to mitigate their impact and optimise the benefits of vaccination programs for individuals and society as a whole.

## 2. Materials and Methods

We adhered to the STROBE Statement checklist for cross-sectional studies [16] for reporting.

### 2.1. Study Design, Setting and Participants

The QoVAX Statewide Study was registered on the Australian New Zealand Clinical Trials Registry (ACTRN12622000020785) [17] and the Health Studies Australian National Data Asset (HeSANDA) collections (https://doi.org/10.60540/XYC4ZN, accessed on 15 June 2024). The pre-specified health economic analysis plan can be found in Appendix A. The primary objective of the overall study was to evaluate serum Spike IgG concentrations in persons who had received at least one COVID-19 vaccine in the 12 months prior to enrolment. Recruitment occurred with electronic consent and data collection via a public-facing website from 27 June to 30 September 2022. This online method was selected due to ongoing COVID-19 restrictions at the time, which limited in-person research activities, and to facilitate rapid, large-scale recruitment across Queensland. Approximately 1 million email invitations were distributed ranging from 5000 to 90,000 per day to eligible participants registered on Queensland’s COVID Vaccine Management System, which is a Queensland Health register recording all COVID-19 vaccines administered in publicly funded COVID-19 vaccination centres. In addition, a wide recruitment campaign was facilitated by media, radio, television, online stories, and social media. Outreach recruitment clinics also occurred in settings such as remote communities, residential care and retirement settings and “pop-up clinics” in some Aboriginal and Torres Strait Islander communities. Using a digitally-delivered survey, the study was open to all people aged ≥18 years residing in Queensland regardless of vaccination status [15]. Individuals were unable to participate if they were unable to decline or give informed consent; did not have access to a smart phone or computer to read the participant information and complete the consent form and questionnaire and had insufficient English language skills. For this analysis of AEFIs, participants were excluded if they had not received any COVID-19 vaccine within 12 months prior to enrolment.

### 2.2. Data Collection and Outcomes

Consenting participants self-completed an online questionnaire using an electronic device, smartphone or computer or, in some settings (e.g., people at indigenous communities or residential care facilities stated a preference to do so, or internet was not available), paper-based forms either self-completed or completed by research staff in individual interviews. Support from centrally located research staff was available by telephone if participants had difficulty navigating the online data collection form.

The questionnaire collected extensive demographic and epidemiological data and asked about the timing and type of vaccination doses received as well as recollection of local and systemic symptoms experienced after each dose, duration of symptoms (≤24 h, 1–3 days, 4–7 days, >1 week), and any resulting healthcare use and time taken off work. Within the questionnaire, a list of five local symptoms (*heat*, *swelling*, *pain*, *itch*, *tenderness*) and 10 systemic symptoms (*fever*, *nausea*, *shivering*, *chill*, *headache*, *tiredness*, *muscle ache*, *joint pain*, *clot*, *other*) could be selected; these were included to ensure consistency with other large COVID-19 vaccination surveys including the World Health Organisation COVID-19 research questionnaire [18] and the AusVaxTracker questionnaire [12]. There was also an option for respondents to select ‘other’ with a free text field to describe the symptom.

The questionnaire also collected information about general health, social and well-being in relation to demographics; family medical history; general health and well-being; health-related quality of life (using the validated EQ-5D-5L questionnaire [19]); absenteeism days; and medication use. Participant data were subsequently linked to routinely collected Queensland Health databases to identify any emergency department (ED) presentations, hospital admissions and non-admitted outpatient occasions of service occurring within a 12-month period. In this study, we investigated outcomes that described self-reported AEFIs, healthcare resource use and absenteeism days following COVID-19 vaccination.

### 2.3. Data Analysis

Descriptive statistics were applied to summarise participant characteristics (age, gender, BMI, Aboriginal and Torres Strait Islander status, educational level, labour force status, number of chronic conditions [0, 1, or >1], and EQ-5D-5L utility score), vaccine doses, AEFIs, healthcare service usage, and absenteeism days. The presence of pre-existing chronic conditions was also collected and summarised in Appendix A. The EQ-5D-5L utility scores were calculated using the latest EQ-5D-5L value set for Australia [20]. Heat maps were plotted to show the percentage and duration of AEFIs.

Healthcare costs for self-reported general practitioner (GP), ED, and community clinic nurse or Indigenous Health Worker visits were estimated using national reimbursement rates from the Medicare Benefits Schedule [21]. Self-reported hospital admissions were cross-validated with linked healthcare data to confirm admissions were likely related to AEFIs from vaccination. Where discrepancies arose between self-reported and linked data, priority was given to the linked data. Admission costs were assigned by applying national price weights based on diagnosis-related groups (DRGs) and length of stay [22]. Ambulance costs were estimated based on the cost per incident reported in the Queensland Ambulance Service 2022 [23]. Cost for the Queensland 13Health phone service (a free 24-hour telephone service providing health advice) was estimated based on the wage rates as at 2022 for a Registered Nurse in Queensland Health [24].

Indirect (absenteeism) costs refer to the productivity losses incurred due to time off work arising from AEFIs. Absenteeism costs were estimated using the human capital approach [25], which is a widely used method for quantifying the productivity-related costs of illness. This involved multiplying the number of hours absent during the study period with average wage rates, by age group (15–19, 20–24, 25–34, 35–44, 45–54, 55–59, 60–64, 65 and over), gender (males and females) and work category (full time, part time, and others) [25]. Wages were based on the 2022 Queensland Average Weekly Earnings from the Australian Bureau of Statistics (ABS) [26]. The survey-reported absenteeism rates and costs were then extrapolated to the total Queensland adults aged >20 years (*n* = 4,216,822 at September 2024) to estimate the overall economic burden, accounting for the weighting of age/sex/work groupings in the survey versus general population [27].

Subgroup analyses were conducted to investigate outcomes for each dose. Specifically, we also investigated the AEFIs (local and systemic) for those who took dose 1 solely (without subsequent doses), doses 1 and 2 (without doses 3 and 4), doses 1–3 (without dose 4), and doses 1–4. All statistical analyses and plotting were conducted in RStudio version 4.3.1 (e.g., dplyr, tidyverse, lubridate, and ggplot2).

Ethics approval: Ethical approval was granted by Metro North Health, Queensland University of Technology prior to study commencement: (HREC/2021/QRBW/81904).

## 3. Results

The participant selection is displayed in Figure 1. There were 6964 respondents from 86% of Queensland’s geographical areas. Of these, 167 were excluded due to age under 18 years (*n* = 1) and incomplete demographic data (*n* = 166). Of the 6797 eligible survey responders, 6777 (99.7%) had recorded data for at least one dose, 6747 (99.3%) for at least two doses, 6268 (92.2%) for at least three doses, and 3084 (45.4%) for at least four doses.

### 3.1. Participant Demographics and Characteristics

Table 1 provides an overview of the demographic characteristics of the participants at the time of the survey. Participants had a mean age of 53.1 ± 14.5 years, with a median of 54.2 years; 67.8% were females with 2.2% identifying as Aboriginal and/or Torres Strait Islander. The mean BMI was 28 kg/m^2^ with one third being obese. Tertiary education (with bachelor’s degree or higher) was attained by one third, 45.0% were in full-time employment, and 17.9% were in part-time roles. Chronic conditions were reported in 40% with 17% reporting two or more. Quality of life at time of enrolment, as measured by EQ-5D-5L utility scores, was 0.94. Notably, participants across doses had similar demographics, but the dose 4 cohort was older and had fewer full-time workers. Detailed cohort characteristics for each dose were reported in the Appendix A.

### 3.2. Vaccine Type and AEFIs by Dose

The BNT 162b2 mRNA vaccine was the predominant vaccine administered across all doses (Figure 1 and Appendix A). This was followed by the ChAdOx1 adenoviral vector for doses 1 and 2 and mRNA-1273 for doses 3 and 4. Experiencing any AEFIs were reported by 3617 (53.4%) individuals after dose 1, 2978 (44.1%) after dose 2, 2550 (40.7%) after dose 3, and 1262 (40.9%) after dose 4 (Appendix A).

A decreasing rate of reported local and systemic AEFIs from dose 1 to dose 4 was observed (Appendix A). The overall percentage of participants with any local AEFIs for dose 1, dose 2, dose 3, and dose 4 was 46.5%, 39.2%, 36.3%, 37.3%, respectively, and the percentage for systemic AEFIs was 42.7%, 34.0%, 29.0%, 28.1%, respectively (Appendix A).

Overall, pain and tenderness were the most commonly reported local AEFIs with tiredness and headaches being the most commonly reported systemic AEFIs (Figure 2 and Appendix A). Most COVID-19 vaccine-related AEFIs, particularly local AEFIs, were observed to resolve within three days (Figure 2).

### 3.3. Costs of AEFIs

Overall, only a small proportion of participants vaccinated reporting seeking healthcare assistance: 3.7% (*n* = 252) in dose 1, 3.2% (*n* = 218) in dose 2, 2.1% (*n* = 129) in dose 3, and 0.8% (*n* = 24) in dose 4. Within the subgroup experiencing AEFIs, these proportions were: 7.0%, 7.3%, 5.1%, and 1.9% respectively (Table 2). Participants who sought medical help for AEFIs mostly required GP and ED visits with a relatively small proportion requiring hospital admission and ambulance services. The overall healthcare cost was AUD 87,032 for Dose 1, AUD 261,125 for Dose 2, AUD 57,064 for Dose 3, and AUD 5304 for Dose 4, with an overall cost of AUD 410,525. The mean healthcare cost per person vaccinated was AUD 13, AUD 39, AUD 9, and AUD 2, respectively (Table 2 and Appendix A). However, for those who sought medical care to manage their AEFIs, the average cost per person was AUD 24, AUD 88, AUD 22, and AUD 4, respectively.

For those who reported vaccination-related AEFIs, absenteeism was recorded in 16.5% (*n* = 597), 18.2% (*n* = 542), 15.2% (*n* = 388), and 11.2% (*n* = 141) after each dose. Within this group, there was a mean of 4.7, 7.4, 3.6 and 2.1 absenteeism days per person, respectively (Table 3 and Appendix A). The overall absenteeism cost was AUD 880,233 for dose 1, AUD 1,277,451 for dose 2, AUD 432,663 for dose 3, and AUD 95,222 for dose 4, with an overall cost of AUD 2,685,569. The mean absenteeism cost per person vaccinated within the full sample cohort was AUD 130, AUD 189, AUD 69, AUD 31, respectively (Table 3 and Appendix A). However, for those who reported AEFIs, the mean individual cost for absenteeism was AUD 1494, AUD 2388, AUD 1136, and AUD 690, respectively. When reported absenteeism rates and costs were extrapolated to the total Queensland adult population of 4.21 million [27], the absenteeism costs for the assumed population amounted to approximately AUD 547,703,266 for dose 1, AUD 798,396,840 for dose 2, AUD 291,075,692 for dose 3, and AUD 130,199,288 for dose 4 with an overall cost of AUD 1.77 billion.

## 4. Discussion

### 4.1. Overall Summary of Findings

This study offers a rigorous examination of the AEFIs in the Queensland QoVAX Statewide study, and it represents the first in-depth analysis of the associated economic burden of AEFIs. This study’s key insights reveal a relatively high incidence of self-reported AEFIs post COVID-19 vaccination; however, these were largely mild and short-lasting with only a small portion of affected individuals seeking medical assistance. Notably, the productivity costs attributable to these AEFIs was substantially greater than the direct healthcare expenses incurred.

### 4.2. Integration with Existing Literature

Our findings on the self-reported rates and nature of AEFI are consistent with existing literature demonstrating the overall high levels of safety and efficacy associated with COVID-19 vaccines, wherein AEFIs are chiefly self-limiting. Several systematic review and meta-analysis studies [28,29] regarding the efficacy and safety of COVID-19 vaccines have been recently published. A systematic review of 11 clinical trials has reported that most of the reactions reported were mild to moderate and were resolved within 3–4 days [28]. Other review studies have shown similar results and reported pain, fatigue, and headache as the most common adverse events, which is consistent with our study [30,31,32]. A commentary in *Age and Ageing* revealed that while mild to moderate severity adverse events occur frequently, serious adverse events are very rare [33].

Consistent with prior studies, including the AusVaxSafety surveillance reports [12,13,34], our study confirmed that health service use due to AEFIs was minimal (about 1%); however, a more notable proportion of respondents reported absence from work with up to 20.3% reporting time off work following vaccination [12,13]. We also found a general trend of decreasing rates of healthcare use and productivity losses over time from dose 1 to dose 4. Our investigation extends existing evidence, emphasising that indirect costs from lost productivity are likely to outweigh the direct healthcare expenditures, and for the first time, it puts a monetary value to these costs. While absenteeism costs associated with AEFIs may seem substantial, they should be interpreted within the Australian employment context, where full-time employees are entitled to 10 days of paid sick or carer’s leave per year under the National Employment Standards. Within this entitlement, the financial impact is typically borne by employers (or by the government in the case of public sector employees). However, if an employee’s time off exceeds these limits, the burden may shift to the individuals, through unpaid leave, or it may continue to affect employers due to prolonged workforce disruptions. This context helps explain the absenteeism rates observed in our study and provides important insight into the broader economic implications of vaccination-related AEFIs.

Queensland’s gross state product (GSP) was approximately AUD 503.35 billion in 2022–23 [35], and the state’s total annual healthcare expenditure exceeded AUD 52.46 billion [36] during the same period. While the projected absenteeism-related costs associated with AEFIs (*totally AUD 1.77 billion for the adult Queensland population*) represent a small fraction of these figures, they highlight a potentially avoidable productivity loss particularly during the period of economic recovery from the COVID-19 pandemic. Moreover, these costs may be disproportionately felt within essential workforce sectors, such as healthcare and education, where staff absences can have cascading effects on service delivery. Incorporating these indirect costs into pandemic preparedness planning may help policymakers design more resilient and equitable vaccination rollout strategies.

In the broader Australian vaccination context, parallels can be drawn with influenza programs. A targeted review by the Australian Technical Advisory Group on Immunisation (ATAGI) [37] highlighted that influenza vaccination substantially reduces the healthcare system’s burden while preventing high rates of morbidity and mortality, particularly among at-risk populations. The review emphasises that vaccination remains the most effective strategy to manage respiratory diseases like influenza and mitigate associated economic impacts. Similarly, insights from a discrete choice experiment on influenza vaccines revealed that Australians highly value vaccine attributes such as increased protection and a match to circulating virus strains with a willingness to pay (WTP) up to AUD 32.37 for a match to circulating strains in a severe season and AUD 2.18 per additional percentage of protection [38]. The literature consistently indicates that the public health and economic benefits of widespread vaccination—such as preventing hospital admissions, reducing morbidity, and preserving productivity—far outweigh the short-term costs. By situating our findings within this broader Australian vaccination context, we reinforce that the short-term costs linked to COVID-19 vaccine AEFIs should be considered alongside the substantial benefits of preventing severe COVID-19 outcomes and alleviating healthcare strain.

### 4.3. Implications for Policy/Practice

The findings of this study should be interpreted in light of the known benefits of vaccination. AEFIs were mostly minor when compared to the potential benefits of vaccination in preventing severe illness, hospitalisations, and deaths. As such, the overall economic burden of AEFIs is expected to be modest in comparison to the wider societal benefits of widespread vaccination programs.

To date, there is limited published evidence on targeted interventions to reduce absenteeism following mild AEFIs. However, some organisational policies have been proposed: effectively communicating vaccine-related policies and expectations [39], offering flexible scheduling options for vaccination [40], providing support for employees experiencing vaccine AEFIs, and continuing to promote a safe and flexible work environment [41]. Moreover, the ongoing surveillance of AEFIs is vital to maintain public confidence in vaccination programs and should be integrated into public health strategies to ensure the continued success of vaccination campaigns.

### 4.4. Future Directions

Future studies should focus on quantifying the broader, long-term economic benefits generated by COVID-19 vaccination programs to provide a comprehensive understanding of their impact. These studies should consider various factors, including healthcare cost savings, productivity gains, restored economic activities, reduced burden on social welfare systems, and the revival of international trade and tourism. Such studies would provide a more holistic view of the benefits of vaccination programs and inform policy decisions on resource allocation and health system preparedness for future pandemics.

### 4.5. Strengths/Limitations

This study goes beyond the clinical safety and efficacy of vaccines to assess the economic implications of AEFIs. By examining both direct healthcare costs and indirect productivity losses, it provides a comprehensive view of the associated economic burden. The study included a large number of participants (6797) across a broad geographic area, covering 86% of Queensland post codes. This extensive coverage provides a robust dataset and enhances the potential generalisability of the findings within the region. Unlike most existing studies that focus on a single vaccine dose [12,13,42], our research evaluated the AEFIs across four doses of COVID-19 vaccines. This multi-dose approach allows for a more detailed analysis of the safety and economic impact over the entire vaccination trajectory, providing valuable insights into the varying effects of successive doses, and notably that AEFIs tended to reduce with dose number.

This study has several limitations that should be considered when interpreting the findings. First, the reliance on self-reported data includes the potential for selection bias (i.e., limited data generalisability). The study population exhibited a demographic skew towards middle-aged and older females with chronic conditions as compared to the general population of Queensland (mean age of 39.1 and 50.5% female) and Australia (mean age of 39.3 and 50.4% female) [43]. This overrepresentation may reflect a greater willingness among this demographic to engage with health-related surveys and report AEFIs, but it may also limit the generalisability of the findings to other groups.

Second, there may have been recall bias particularly given the retrospective nature of data collection. Participants were asked to recall adverse events and work absences following multiple vaccine doses over a period of up to 12 months, which may have resulted in underreporting or the misclassification of mild or short-lived symptoms. While we were able to cross-validate linked hospital admissions data, it was not possible to verify other self-reported healthcare use including GP visits and ambulance transfers. Future studies could benefit from obtaining routinely collected data to validate these observations.

Third, the use of online data collection methods, while appropriate under pandemic-related constraints, may have introduced participation bias. Individuals who experienced AEFIs may have been more motivated to participate in the survey, potentially overestimating the true prevalence and impact of AEFIs in the vaccinated population. In addition, those without internet access or with lower digital literacy may have been underrepresented. Future research could compare online and in-person recruitment methods to better understand their respective strengths and limitations in capturing AEFI data, thereby enhancing the generalisability and robustness of findings in this research area.

Fourth, this study focused on estimating the overall health economic burden of AEFIs at the population level; we acknowledge that the economic impact may vary by demographic characteristics such as age, occupation type, or the presence of comorbidities. Future research should explore these subgroup-specific effects to inform more targeted public health strategies and ensure equitable resource allocation during mass vaccination efforts.

Finally, the predominance of the BNT 162b2 mRNA vaccine in our cohort may not fully capture the AEFI profiles of other vaccines used during the COVID-19 vaccination campaign. Exploring the adverse event profiles of other vaccine types would provide a more comprehensive understanding of vaccine safety.

## 5. Conclusions

This study provides novel evidence on the economic burden of AEFIs following COVID-19 vaccination in Australia, highlighting that the indirect costs due to productivity losses far outweighed the direct healthcare-related expenses. Whilst modest in relation to the broader economic landscape, the short-term productivity costs should be acknowledged in future pandemic planning and occupational health strategies. A comprehensive understanding of the direct and indirect impacts of AEFIs is essential to inform public health messaging, guide policy development, and ensure effective resource planning in large-scale immunisation programs.

## Figures and Tables

**Figure 1 vaccines-13-00712-f001:**
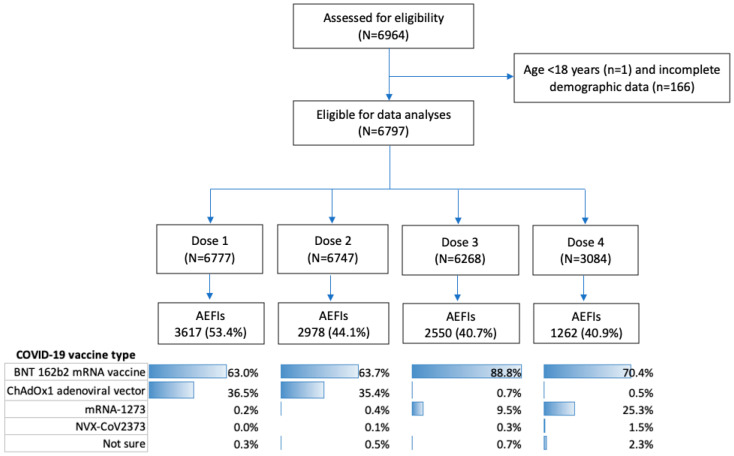
Flow chart of participant selection for data analysis.

**Figure 2 vaccines-13-00712-f002:**
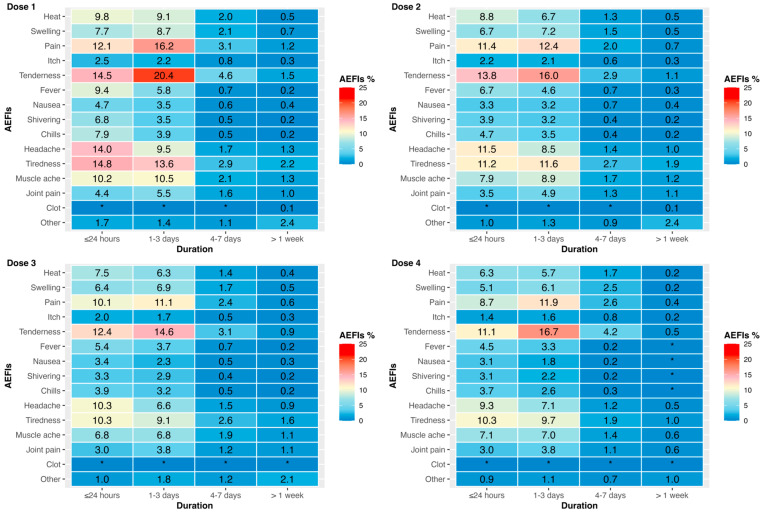
Percentage of participants reporting each adverse effect for each dose by duration. The AEFIs included local (heat, swelling, pain, itch, tenderness), systemic (fever, nausea, shivering, chill, headache, tiredness, muscle ache, joint pain, clot), and other symptoms. Asterisk (*) signals indicate the values < 0.1%, equal to the participant number of <5. Abbreviation: AEFIs, adverse effects following immunisation.

**Table 1 vaccines-13-00712-t001:** Participants’ characteristics at the time of survey (overall cohort).

	Overall Cohort (N = 6797)	
Variable	Value	%
**Age**, years		
Mean (SD)	53.1 (14.5)	NA
Median	54.2 (42.2, 64.0)	NA
Min, Max	[18.0, 96.5]	NA
**Gender**		
Male	2183	32.1%
Female	4606	67.8%
Missing	8	0.1%
**Aboriginal and/or Torres Strait Islander**		
Yes	149	2.2%
No	6531	96.1%
Missing	117	1.7%
**BMI**, years		
Mean (SD)	28.0 (6.3)	NA
Median	26.9 (23.7, 31.2)	NA
Min, Max	[15.4, 94.3]	NA
**Obesity**		
Yes (BMI > 30 kg/m^2^)	2057	30.3%
No	4584	67.4%
Missing	156	2.3%
**Education**		
High school or below	1206	17.7%
Certificate/diploma degree	1881	27.7%
Bachelor degree	1645	24.2%
Postgraduates or above	739	10.9%
Missing	1326	19.5%
**Labour force status ^#^**		
Full time	3027	44.5%
Part time	1207	17.8%
Casual	423	6.2%
Unemployed	161	2.4%
Not in labour force (this includes carers, volunteers, students, retirees, and home duties)	185	2.7%
Other	1698	25.0%
Missing	96	1.4%
**Chronic conditions**		
0	4063	59.8%
1	1595	23.5%
>1	1139	16.7%
**EQ-5D-5L ***		
Mean (SD)	0.94 (0.11)	NA
Median	0.96 (0.92, 1.00)	NA

^#^ ABS definition for full-time, part-time, casual, and unemployment: https://www.abs.gov.au/statistics/detailed-methodology-information/concepts-sources-methods/labour-statistics-concepts-sources-and-methods/2023/concepts-and-sources/employment-arrangements#status-in-employment; access on 5 July 2024. * The mean EQ-5D-5L utility scores for Australian general populations aged 45–54 years was 0.89 with a standard deviation of 0.16, and our estimate aligned with this population norm. Abbreviations: BMI, body mass index; SD, standard deviation; NA, not applicable; EQ-5D-5L, EuroQol five-dimension five-level.

**Table 2 vaccines-13-00712-t002:** Self-reported healthcare service use due to COVID-related adverse effects.

	Dose 1		Dose 2		Dose 3		Dose 4	
	Number of Individuals	Average Number of Times	Number of Individuals	Average Number of Times	Number of Individuals	Average Number of Times	Number of Individuals	Average Number of Times
13 Health ^#^	31 (12.3%)	1.26 ± 0.82	20 (9.2%)	2.00 ± 2.75	12 (9.3%)	1.17 ± 0.39	2 (8.3%)	1 ± 0
Ambulance	14 (5.6%)	1.43 ± 0.94	18 (8.3%)	1.28 ± 0.67	6 (4.7%)	1.83 ± 0.98	1 (4.2%)	NA
Community clinic nurse or indigenous health worker	6 (2.4%)	1.00 ± 0.00	5 (2.3%)	1.60 ± 0.89	2 (1.6%)	1.50 ± 0.71	0 (0.0%)	NA
General practitioner (including after-hours/home visit/telephone consult)	135 (53.6%)	2.48 ± 3.24	120 (55.0%)	4.50 ± 10.44	86 (66.7%)	2.86 ± 3.36	20 (83.3%)	1.70 ± 1.30
Emergency department	62 (24.6%)	1.29 ± 0.66	47 (21.6%)	1.68 ± 1.02	21 (16.3%)	1.38 ± 0.74	1 (4.2%)	NA
Admitted to public hospital	4 (1.6%)	1.09 ± 0.30	8 (3.7%)	1.23 ± 0.69	2 (1.6%)	1.29 ± 0.49	0 (0.0%)	NA
**Total number of individuals reported healthcare use**	**252**		**218**		**129**		**24**	
*% among participants reporting* AEFIs	*7.0*% (*252*/*3617*)		*7.3*% (*218*/*2978*)		*5.1*% (*129*/*2550*)		*1.9*% (*24*/*1262*)	
*% among participants* vaccinated	*3.7*% (*252*/*6777*)		*3.2*% (*218*/*6747*)		*2.1*% (*129*/*6268*)		*0.8*% (*24*/*3084*)	
**Overall healthcare use costs** *	**$87,032**		**$261,125**		**$57,064**		**$5304**	
*Healthcare cost per person reporting AEFIs*	*$24*		*$88*		*$22*		*$4*	
*Healthcare cost per person vaccinated *(*overall cohort*)	*$13*		*$39*		*$9*		*$2*	

* Healthcare costs are detailed in Appendix A. ^#^ 13HEALTH (13 43 25 84; https://www.qld.gov.au/health/contacts/advice/13health; accessed on 20 June 2024) is a confidential phone service that provides health advice to Queenslanders.

**Table 3 vaccines-13-00712-t003:** Self-reported absenteeism (days) due to COVID-related adverse effects.

	Dose 1	Dose 2	Dose 3	Dose 4
**Number of individuals who reported absenteeism**	597	542	388	141
*% among participants reporting AEFIs*	*16.5*% (597/*3617*)	*18.2*% (542/*2978*)	*15.2*% (388/*2550*)	*11.2*% (141/*1262*)
*% among participants vaccinated* (overall cohort)	*8.8*% (597/*6777*)	*8.0*% (542/*6747*)	*6.2*% (388/*6268*)	*4.6*% (141/*3084*)
**Absenteeism, days**				
Mean (SD)	4.7 ± 23.7	7.4 ± 37.7	3.6 ± 12.3	2.1 ± 6.7
Median (IQR)	1 (1, 2)	1 (1, 2)	1 (1, 2)	1 (1, 2)
[Min, Max]	[0.2, 365.0]	[0.3, 365.0]	[0.5, 180.0]	[0.1, 80.0]
**Overall absenteeism cost**, AUD	**$880,233**	**$1,277,451**	**$432,663**	**$95,222**
Absenteeism cost per person reporting AEFI *	$1494	$2388	$1136	$690
Absenteeism cost per person *vaccinated* (overall cohort) ^#^	$130	$189	$69	$31

Absenteeism costs are detailed in Appendix A. * Absenteeism cost per person reporting AEFI (=“Overall absenteeism cost”/“number of participants reporting AEFIs”). ^#^ Absenteeism cost per person vaccinated [overall cohort] (=“Overall absenteeism cost”/“number of participants vaccinated [overall cohort]”). Abbreviations: AEFI, adverse events following immunisation; AUD, Australian Dollar; SD, standard deviation; IQR, Interquartile Range.

## Data Availability

To inquire about access to QoVAX data, refer to “Queensland COVID-19 Vaccination (QoVAX) Safety and Efficacy Trial Program: Mixed Dose 1 and 2 Study”, https://doi.org/10.60540/9HWYM8, accessed on 15 June 2024, QCIF Dataverse, V1, HeSANDA Collections.

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
