# Peer review of "Adverse Events and Associated Economic Burden of COVID-19 Vaccination in Queensland, Australia: Findings from the Cross-Sectional QoVAX-Statewide Study"

_vaccines, 2025, doi:10.3390/vaccines13070712_

Round 1
Reviewer 1 Report (New Reviewer)
Comments and Suggestions for Authors
For four COVID-19 vaccination doses in Queensland, Australia, what are the frequency, type, and duration of AEFIs? Kindly clarify in the introduction.
For individuals and the healthcare system, what is the total economic load direct and indirect costs of AEFIs?
In Queensland, how do the economic effects of AEFIs compare to those of other areas or nations having like immunization campaigns?
Do various demographic groupings (e.g., age, occupation, comorbidities) have varying economic burdens?
Line 44: In those who have at least one COVID-19 vaccination over the past 12 months, what are the serum Spike IgG concentrations? Exist demographic or clinical variables (such as age, sex, location, underlying medical disorders) linked to either higher or lower IgG concentrations following vaccination?
Does the type of vaccination (Pfizer, Moderna, AstraZeneca) produce distinct IgG responses or AEFI profiles?
Section 2.2: Comparatively to digital self-entry, how trustworthy are the self-reported symptoms especially those gathered via paper forms or oral interviews?
Given vaccine related symptoms, what percentage of participants needed outpatient treatments, hospital admission, or emergency department treatment?
Section 2.3: Exists any difference between linked healthcare data and self-reported data? Indeed, how were they handled?
Section 3:2
Why does each next dose cause the incidence of AEFIs local and systemic to drop? kindly defend it.
Which demographic groups that is, age, sex, chronic condition status were more likely to report AEFIs or seek medical advice? Please indicated it, if at all possible.
Dose 2 had the highest personal healthcare expense; what kinds of treatments helped to explain this? Any particular reason with dose 2, kindly clarify!!
In the cost and symptom studies, how were vaccination kinds and combinations handled?
How can the findings affect public confidence or public communication policies concerning vaccination safety?
Section 4: under review
How would the results have been affected by selection bias that is, overrepresentation of older women with chronic illnesses?
Given a retroactive recall period of up to 12 months, how trustworthy are the estimates of productivity costs?
Do some vaccine types e.g., BNT162b2 against ChAdOx1 vs. mRNA-1273 show varying safety or cost profiles across doses?
How should legislators balance long-term benefits such as reduced hospitalizations, more stable workforce against short-term absence costs?
In real-time vaccination campaigns, what systems might enable validation and improvement of productivity cost models?
In what ways might mixed methods quantitative plus qualitative help to clarify how AEFIs affect behavior (e.g., absenteeism, vaccine hesitancy)?
How might objective health system data such as Medicare, hospital records be included into future research to validate self-reported AEFIs?
Tell succinctly whether, should another viral pandemic or a similar or more advanced COVID-19 mutation arise in the future, the AEFLs system would remain valid...
Author Response
please see attached file.

Reviewer 2 Report (Previous Reviewer 2)
Comments and Suggestions for Authors
The resubmitted Manuscript titled “Adverse Events and Associated Economic Burden of COVID-19 Vaccination in Queensland, Australia: Findings from the Cross-Sectional QoVAX-Statewide Study” tackles a crucial topic, adverse events following immunization (AEFIs) related to COVID-19 vaccines in Australia. It also delves into the financial impact these AEFIs have on both the healthcare system and the broader national economy.
Since the previous review, the authors have addressed some of the feedback, particularly by improving sections of the abstract, introduction and conclusion. While this marks progress in the right direction, many key comments regarding the methodological approach, the results and their interpretation remain unaddressed. These unresolved issues are critical, and until they are properly tackled, the manuscript is not ready for publication. Also, I have to insist on a point-by-point response to my comments, including the explanation, corrections and the final text that will be implemented in the Manuscript, as this is the second time we are encountering the same problems in the Manuscript.
General comments:
- The Authors still fail to explain the rationale behind their chosen data collection method, as well as its potential implications.
- Aside from noting that respondents came from 86% of Queensland’s geographical areas, the Authors still do not place the study population in the wider context of the vaccinated population in Queensland or Australia. Without this, it is hard to see how the results can be confidently generalized.
- The supplementary material still does not include the questionnaire used, and some results presented in the study, such as the number of sick leave days, do not have a corresponding question in the data collection instrument (let us call it this) to clarify how the data was collected.
- The Materials and Methods section lacks a clear explanation and justification for how the calculation of indirect (absenteeism) costs was conducted. Given that this is one of the main objectives of the study, a well-reasoned and transparent explanation is essential. Without it, readers are left without a clear understanding of the basis or validity of the approach taken.
Materials and Methods:
- The Authors still do not provide basic information or a description of the general population in Queensland that the study sample is meant to represent.
- There is still no explanation of how participants’ data was linked to databases covering ED presentations, hospital admissions, or outpatient occasions of service (Data Collection and Outcomes, Lines 32–35).
- In the Data Analysis section (Lines 45–47), it is stated that ED presentations and outpatient occasions were self-reported, which directly contradicts the earlier implication that these were identified through data linkage.
- Furthermore, in Lines 47–49, the sentence “Self-reported hospital admissions were cross-validated with linked healthcare data to confirm admissions were likely related to AEFIs from vaccination” lacks the necessary detail to understand how this validation process was carried out.
- Data Analysis (Lines 4–10): There is no explanation of what is meant by indirect (absenteeism) costs. The manuscript lacks a definition of the term and does not clarify why these costs were calculated by multiplying the number of hours absent during the study period with the average wage rates by age group, gender, and work category. It is also unclear why this specific method was chosen, and whether it aligns with standard practices for calculating indirect costs in similar research. Since the manuscript repeatedly claims that this is a novel analysis and that similar research has not been done before, this only reinforces the need for a clear, detailed explanation of the methodology used.
- The manuscript also fails to explain the types of work categories, later listed under “Labor force status” in Table 1, particularly “Casual” (6.2%) and “Other” (25.0%). It is not specified what these categories include or how they were defined. Furthermore, it’s unclear which of these categories were included in the calculation of indirect (absenteeism) costs related to AEFIs.
- There is an inconsistency in the reported number of Queensland adults aged over 20. Lines 8–10 state this figure as 4,109,753, while the “Cost of AEFIs” section (Lines 11–14) refers to a total adult population of 3.6 million. This contradiction needs to be resolved.
Results:
- The resolution of Figure 2 remains poor, making it difficult to read and interpret. While Figure 1 is easier to understand, its low resolution also comes across as unprofessional. Both figures should be replaced with high-resolution versions to ensure clarity and improve the overall presentation of the manuscript.
- Figure 1: The horizontal bar charts use inconsistent x-axis scaling. In some cases, bars representing different percentages (e.g. 63.7% and 88.8%) appear nearly identical in length, which can easily mislead readers. This kind of visual distortion affects the clarity and integrity of the data presentation. All bar charts should be revised to use a consistent x-axis scale so that the bar lengths accurately reflect the corresponding values.
- Lines 22–24 state that “Of 6,797 eligible survey responders, 6,777 (99.7%) received at least one dose,” yet the Study Design, Setting and Participants section (Lines 44–48) describes eligibility as including only individuals who had received at least one COVID-19 vaccine dose in the 12 months prior to enrolment. This raises a clear contradiction: if having received at least one dose was an inclusion criterion, it is unclear how 0.3% of eligible participants reportedly had not received a dose. This discrepancy needs clarification.
- There is still no clear explanation in the manuscript or supplementary material regarding how absenteeism days were gathered, or how the mean absenteeism per vaccine dose was calculated. Additionally, the distribution of this data is not presented. The presence of extreme values (e.g., 365 days for doses 1 and 2, 180 and 80 days for doses 3 and 4) could have significantly skewed the mean, especially given that the reported median is 1 day for all doses. This raises concerns about the validity of the average values used in the cost calculations and must be addressed transparently.
- Furthermore, the manuscript still does not explain why the specific method used to calculate absenteeism-related costs was chosen, or why it is appropriate for this type of analysis. Given that the authors emphasize the novelty of the study, it is especially important to justify this methodological choice.
- Finally, while the estimated absenteeism costs extrapolated to the adult population of Queensland are described as being in the hundreds of millions of AUD, no contextual information is provided to help the reader interpret the significance of these figures. For example, how do these costs compare to pre-pandemic or post-vaccination economic indicators? Were there any notable strains on the healthcare or economic systems at the time? Without this context, the figures lack interpretability and impact.
Discussion:
The Discussion section places heavy emphasis on the significance of costs associated with productivity loss due to absenteeism from AEFIs. However, this argument is difficult to follow for several reasons.
- The term indirect (absenteeism) costs is never clearly defined.
- There is no explanation of how the data on mean absenteeism days was collected or calculated.
- The rationale for the cost calculation method is not provided, nor is the method itself adequately described.
- It is unclear which stakeholder (e.g. individual, employer, healthcare system, or government) is bearing the financial burden of these losses.
- The monetary values presented lack any contextual framing — there is no reference to economic indicators, baseline productivity costs, or historical data to help the reader assess the impact.
Without addressing these gaps, the current discussion lacks clarity, depth, and relevance. A substantial rewrite is needed to meaningfully support the argument and ground the findings in proper context.
Conclusions:
The entire Conclusions section should be revised to align with the corrected and clarified content of the manuscript.
Author Response
please see attached file.

Reviewer 3 Report (Previous Reviewer 3)
Comments and Suggestions for Authors
All of my previous comments have adequately been addressed and I have no further recommendations.
Author Response
please see attached file.

This manuscript is a resubmission of an earlier submission. The following is a list of the peer review reports and author responses from that submission.
Round 1
Reviewer 1 Report
Comments and Suggestions for Authors
This is a well-conducted, well-communicated study that is relevant from a health management perspective.
Reviewer 2 Report
Comments and Suggestions for Authors
The Manuscript „Adverse Events and Associated Economic Burden of COVID-19 Vaccination in Queensland, Australia: findings from the cross-sectional QoVAX-Statewide study“ explores an important topic of adverse events following immunization (AEFIs) connected to COVID-19 vaccination in Australia. The Authors have used an online survey with convenience sampling, and have described the most frequent AEFIs and their healthcare and productivity costs.
The Manuscript is well-written and deals with an important topic connected to any vaccination campaign, or even any health intervention. Both the Introduction and the Discussion are very general, and do not provide enough detail of the previous studies, actual findings, and how this study closes some knowledge gaps. Sometimes, the Authors provide statements followed by references, where it is unclear if the statement is their own result, or their interpretation of the result, or something other authors have found (but then some context and specific info should be provided). The Methods section does not allow, in its current form, reproducibility of the study, and the Authors should make sure all of the relevant data collection instruments and calculations are explained in detail so that other authors could apply the methodology in their settings. The use of heatmaps to visualize data is original and useful, with minor modifications (see below). I suggest doing some analytical statistics and reliability analysis to confirm the findings, and a stronger Discussion of the results, comparing to similar studies in Australia but also internationally (helping generalize the results) could benefit the readers, and could increase the impact of this well-done study. After those modifications, I would strongly recommend the publication of this Manuscript.
General comments
The Manuscript uses a structure which might not be according to Journal standards (e.g. „Summary“). Please check.
Although methodologically sound, the Authors should focus more attention on explaining their choice and implications of data collection method (e.g. would most serious AEFIs be missed, what is the potential target population’s use of the recruitment channels used in this study, would the results be any different if this study were to be done in person at primary healthcare facilities, etc).
To better understand the implications of this research, the Authors should discuss some other common diseases and/or vaccinations and compare the costs (healthcare and productivity) using the same standards, and what was the cost of getting COVID-19 – even if just calculating using statistical data for Australia (e.g. average length of stay in hospital, etc...).
How does the study population match the general population of Australia or Queensland, or the population which was vaccinated in Australia?
Since various questions used in this study come from different study protocols and questionnaires, to enable reproducibility – I strongly suggest that the Authors provide an easy-to-use questionnaire which includes all the data they were collecting for this study, or provide the protocols/questionnaires publicly available at some repository.
Abstract
Are the subsections in the Abstract required by the Journal and do they match the requirements? Maybe the abstract should have a more narrative form.
AEFI – please define on first use.
Please provide more detail as to the study population recruitment channels, and some basic description of the study population (age, gender, work status, etc...).
Summary
I do not find this section any more informative or useful than the Abstract. The Authors might think about making it more informative and to the point, IF there is anything to mention here that is not already in the Abstract.
Line 42 – it is safe to say „mild to severe“ AEFI. Or it would be more informative to provide a % or a number here.
Line 43 – this part should focus only on the results (new), and not repeat the Methods, if you decide to keep the Summary.
Introduction
The Introduction is too general in the most part, and it would be really helpfull if the Authors can provide some up-to-date data on the AEFIs worldwide (statistics) and in Australia according to official agencies, and some previous research on the topic.
Line 61 – not all AEFIs are mild, and here it would be great to better understand the „known“ – what are the most common AEFIs? Which are mild, severe, etc...? How often do they happen? Any previous research in Australia or what does the official data say?
Line 62-64: the Authors have not shown that „...warrants further investigation“. Where is the knowledge gap? You start to examine this only later, so there is not base to say further investigation is needed at this point.
Lines 65-74: please provide actual data on efficacy and safety of COVID-19 vaccines, the frequency of AEFIs, and when you discuss the healthcare utilization – show the results of research you are building upon.
Where do definitions in Supplementary 1 come from?
Lines 78-81: how about worldwide studies? Any info there?
Methods
Line 93: where is the „pre-defined economic evaluation plan“ available? This would be important for reproducibility and repeatability purposes. Is this in the HeSANDA collections? I was not able to access it.
Lines 102-106: please provide more information about the recruitment process. E.g. how big is the email database that you used, does it include all people vaccinated in Queensland. Here, it might be useful to give some basic info about Queensland, the population, etc... I was also unable to find where the list of local and systemic AEFIs comes from.
Lines 136-139: How was the participant data linked to ED presentations?
The „Analysis“ should be a section within „Methods“. This section should include detailed information about the statistical analyses, including R packages that you used.
It seems there are no analytical methods used, only descriptive. I believe it would be beneficiary to try to identify some factors associated with higher cost (other than employment status) and higher risk of AEFIs.
Lines 150-165: I believe this part should be before the statistical analysis in a subsection of the „Methods“. Make sure all the necessary information to enable reproducibility are here.
Could a person have received more than 1 dose in the last 12 months? How was this handled?
Results
Figures should be of higher quality/resolution.
Line 174: what does it mean from „86% of Queensland post-codes“?
Line 188: What does a QoL of 0.94 mean? Can you put this in a context?
Figure 2: The coloring should all be on the same scale, e.g. Dose 1 RED is 20% while Dose 2 RED is max 16%.
Line 213: You explain the asterisk as less than 1%, but in some fields (Figure 2) you show 0.2%, while the asterisk is there only for clots.
Table 2 and Supplementary Table 2: What is „13 health“?
Lines 227-237: Could you explain in the Methods part what is (and how it is calculated) the mean absenteeism cost per person, mean individual cost (and who is at lost – how much does a person loose, how much the employer, how much the insurance or who else might be involved), and how is the absenteeism cost extrapolated to the total Queensland adult population.
Considering the average AEFI cost and absenteeism days per person, it is obvious that the AEFIs reported were mild. On the other hand, the high absenteeism cost either says that people decide to be absent although they have a mild AEFI (e.g. in some cultures it would be unacceptable to be absent from work due to redness or pain in the arm after a vaccine), or that the salary standards in Australia are very high. This should be discussed in the Discussion. For example, should someone recommend, due to high absenteeism costs that only retired persons should get vaccinated in Australia, and those at high risk (comorbidities) – what would the cost be then?
Table 3: I notice a huge range of days absent in each group, with a median of 1, and Max of 365 for Doses 1 and 2, and 180 for dose 3, and 80 for dose 4. Could analysis of the influence of extreme values or reliability analysis help estimate how your estimates behave in different settings?
Would it be useful to include confidence intervals for all the estimates (number and % of AEFIs, number and % of those reporting absenteeism, days of sick leave, etc...)?
Discussion
The Discussion, similar to the Introduction, does not go into enough detail to analyze and compare specific results, and to try to explain them. Also, it is focused on the Australian settings, or quotes „similar findings“ without going into details about where the similarities are and where and why there might be differences.
Have there been any interventions to reduce the absenteeism due to mild AEFIs after vaccination in the literature? It seems, and this comment might come from my cultural background, that in some cultures it is very acceptable and has become a norm to be absent from work due to a mild (and clinically insignificant AEFI), which drives the economic burden of vaccination up. This could drive vaccine hesitance in the future, and you are in a good position to address this in your discussion.
Conclusions
The Conclusions section should not repeat results, but should focus on the main findings and what the implications of those findings could be. The study is very well done, and the results are interesting, but the implications should go beyond re-stating the results.
Reviewer 3 Report
Comments and Suggestions for Authors
- Abstract: The abbreviation AEFI has not been introduced.
- One major limitation is the online recruitment. This may lead to the bias that people with AEFI are more willing to participate in the study.
- Furthermore, calculating the economic effects of AEFI misses the comparison of economic effects of severe COVID-19 cases which might have been prevented due to the vaccination.
- The whole information has been assessed retrospectively. This may be totally bias when people have received vaccinations at three times or even more.
- Methods: It is not clear why in some cases information were collected paper-based.
- Chapter 4 should be a sub-chapter of the methods section.
- Results: You mention that 99.7% received at least one dose. However, in the inclusion criteria you said that only persons with at least one dose have been included.
- The discussion is much too short and needs to be improves substantially, by critically discussing the implications of the study and comparing the results with previous studies.
